# Relationship between Protein Digestibility and the Proteolysis of Legume Proteins during Seed Germination

**DOI:** 10.3390/molecules28073204

**Published:** 2023-04-04

**Authors:** Indrani Bera, Michael O’Sullivan, Darragh Flynn, Denis C. Shields

**Affiliations:** 1Conway Institute of Biomolecular and Biomedical Research, University College Dublin, D04 V1W8 Dublin, Ireland; 2School of Medicine, University College Dublin, D04 V1W8 Dublin, Ireland; 3UCD Institute of Food and Health, School of Agriculture and Food Science, University College Dublin, D04 V1W8 Dublin, Ireland; 4Flynn & Flynn Global Trade Ltd., T/A The Happy Pear, A67 EC56 Wicklow, Ireland

**Keywords:** plant proteins, legumes, protein digestibility, germination, peptidomics, proteomics

## Abstract

Legume seed protein is an important source of nutrition, but generally it is less digestible than animal protein. Poor protein digestibility in legume seeds and seedlings may partly reflect defenses against herbivores. Protein changes during germination typically increase proteolysis and digestibility, by lowering the levels of anti-nutrient protease inhibitors, activating proteases, and breaking down storage proteins (including allergens). Germinating legume sprouts also show striking increases in free amino acids (especially asparagine), but their roles in host defense or other processes are not known. While the net effect of germination is generally to increase the digestibility of legume seed proteins, the extent of improvement in digestibility is species- and strain-dependent. Further research is needed to highlight which changes contribute most to improved digestibility of sprouted seeds. Such knowledge could guide the selection of varieties that are more digestible and also guide the development of food preparations that are more digestible, potentially combining germination with other factors altering digestibility, such as heating and fermentation. Techniques to characterize the shifts in protein make-up, activity and degradation during germination need to draw on traditional analytical approaches, complemented by proteomic and peptidomic analysis of mass spectrometry-identified peptide breakdown products.

## 1. Introduction

A global shift from animal-based diets towards more plant-based diets is recommended on both environmental and health grounds [1]. Certain seeds, especially legumes, are high in protein content (20–45%) [2], and their production can lead to less greenhouse emissions, water use, and land use compared with animal proteins, leading to an increased focus on plant protein sources [3,4]. Plants contribute many nutritional benefits in addition to their protein content, with higher unsaturated fatty acids, lower cholesterol content, as well as higher dietary fiber, phyto-nutrients and antioxidants [5]. After grains, legumes from the Leguminosae and Fabaceae families represent a major component of plant foods, and complement them nutritionally in terms of amino acid composition [6]. Various legume species are consumed as a part of human diet worldwide. In a typical plant-based diet, they can provide around 33% of dietary protein, alongside the carbohydrates, fiber, vitamins and minerals that they also provide [7,8]. Major legume seeds used for human consumption are peas (*Pisum sativum* L.), different varieties of beans (*Phaseolus vulgaris* L., *Vigna unguiculata* L., *Phaseolus lunatus* L., *Vicia faba* L.), chickpea (*Cicer arietinum* L.), lentil (*Lens culinaris* Medik.) and lupin (*Lupinus albus* L.) [9,10,11]. They are valued as alternatives to meat-based proteins and are the most important food source worldwide after cereals [12]. Consuming legumes can help in overcoming protein-related malnutrition, thus helping to overcome undernourishment of growing human populations [12]. Other than the presence of macronutrients, legumes contain bioactive compounds which have therapeutic properties [13], and a legume-rich diet appears to have an appreciable impact on heart disease and cancer risk [14,15,16]. When consumed as part of a diet, along with cereals, legumes provide equivalent protein nutrition to a meat- and dairy-based diet [17].

However, differences between animal and plant protein sources, in terms of their digestibility and nutritional quality, are frequently highlighted as a potential limitation of plant-based diets. Plant proteins are often less completely digested than animal proteins [18]. Plant seed proteins are likely to have evolved to be difficult to break down before germination to make them less attractive to microbes, insects, and other herbivores that may be attracted to them as a food source; they have also evolved to allow the protein to be broken down and made available to the seedling during germination. One feature of the seeds contributing to lower digestibility is the presence of protease inhibitors, whose levels may particularly increase in pathogen-exposed plants [19,20]. Another feature is the high protein concentration and low water content, which may create a physical barrier reducing access of digestive proteases to the storage proteins. A third feature is the inherent structures of the storage proteins themselves, including glycosylation and oligomerizations (which may further increase with heating) [21], which may also impede protease access to the protein chain. However, the precise relationship between protein structure and digestibility in legume proteins is incompletely understood [22]. It has been postulated that the higher beta-sheet content of many plant proteins may be associated with poorer digestibility, particularly in heated foods, perhaps because heating induces intermolecular beta-sheet association, resulting in oligomerization into aggregates [21,23].

## 2. Plant Proteins and Their Alterations during Germination

Germination is an important physiological process in plant development. Germination occurs when seed is put in proper physiological context, including the availability of water, oxygen, and appropriate temperature conditions. The process of germination can be divided into three steps. The first step involves (i) imbibition, which involves initial absorption of water to hydrate the seed and (ii) activation of metabolism, with increased respiration and protein synthesis. Imbibition of water makes the seed coats more permeable to oxygen and water. The uptake of water is immediately followed by an increase in respiration, followed by the mobilization of stored reserves including proteins, carbohydrates, lipids, and nucleic acids. Thus, the embryo cells resume metabolic activities for growth, while stored food reserves are mobilized and digested by using the energy generated from aerobic respiration.

Seed proteins can be roughly grouped into three types: major storage proteins, proteases, and protease inhibitors. Since protease inhibitors can inhibit the action of human digestive proteases, they are sometimes considered antinutrients [20]. The digestion of food proteins depends on the interplay of these three components. During germination, proteases break down the storage proteins to provide free amino acids and small peptides [24], which can contribute to the synthesis of structural and functional proteins of the developing radicle. It has been postulated that plants and insects are in evolutionary conflict, with plant protease inhibitors acting as a defense against herbivores that adapt to cope with these barriers to proteolysis [25,26,27]. Protease inhibitors may comprise up to 10% of plant protein content [28]. In plant seeds and pulses, proteases digest substrate storage proteins on germination, and plant seed protease inhibitors may play roles both in inhibiting proteolysis pre-germination, and in inhibiting proteolysis by insects or other herbivores. Protease inhibitors can be both proteins themselves, as well as non-protein inhibitors such as phytates [29]. In the following sections, we review qualitative and quantitative changes in legume seed proteins during germination.

### 2.1. Changes in Crude Protein Content during Germination

Since legumes are the most important sources of plant proteins, changes in protein content of legumes after germination have drawn a lot of attention. Germination is potentially an inexpensive technique to improve the nutritional quality of legumes and other grains. Protein content is typically measured as the nitrogen content, assuming these two measures are approximately equivalent. However, non-protein nitrogen (in chlorophyll, free amino acids, nucleic acids, and other compounds) in legumes may be interpreted as being minor (<10%) or much more substantial, depending on the extraction method used to precipitate protein, and may also vary among pea strains [30,31]. Most of the studies referred to in this review used the Kjeldahl method to estimate protein content, so that, despite the fact that the absolute levels of protein may not be accurate, the estimates do serve as a useful relative indicator of differences in likely protein levels among samples.

Table 1 summarizes the observation across many legumes that germination typically results in an increase in the dry weight percentage of protein. Increase in protein content may well reflect both the breakdown of fats and carbohydrates, and the de novo synthesis of protein and free amino acids [32].

### 2.2. Changes in Polypeptide Molecular Weight Distributions during Germination

Mammila et al. investigated the effects of germination on the molecular weight distribution of proteins for various legumes [49]. As shown in Figure 1, germination increased the concentrations of low molecular weight peptides and decreased the concentrations of higher molecular weight proteins/peptides for all the legumes studied except for kidney bean. These differences are due to different types of proteases and their time of release during the germination process. The morphological and physiological characteristics of different legumes also influences the behavior of proteases [49].

### 2.3. Storage Protein Changes during Germination

The amount of proteins present in seeds varies from species to species, with legumes having the highest quantity (e.g., up to 40% of dry weight), whereas cereals have relatively low quantities (~10%) [50]. The bulk of legume seed proteins are classed as storage proteins. Storage proteins are stored in single membrane-bound organelles known as protein bodies. Osborne in 1924 classified the seed storage proteins into albumins (soluble in water), globulins (soluble in dilute saline), prolamins (soluble in alcohol/water mixtures), and glutelins (soluble in dilute acids or bases) [28]. Of these, globulins are the most important storage proteins of legumes. According to their sedimentation coefficient (S), legume globulins are classified into three sub-categories: 2S, 7S and 11S globulins [51,52] Chickpea consists of 15–30% protein [53]. The major proteins found in chickpea are globulins (53–60%), glutelins (19–25%), albumins (8–12%), and prolamins (3–7%) [54]. Chickpea has a relatively low level of sulfur-containing amino acids [54,55], which could potentially impact on ease of monomer digestibility, including reduced risk of disulphide linkage of oligomers during heating.

While it might be anticipated that soluble proteins should show a decline in their percentage of total protein during germination, in fact, some soluble protein fractions in certain species can increase (Table 2). Together, these findings across legumes suggest that different species and varieties may adopt different strategies, such as activation of different proteases, different rates of proteolysis, and/or alterations to timings of proteolytic events, during germination.

### 2.4. Changes in Free Amino Acids and Protein Amino Acids during Legume Germination

Amino acid composition varies widely among seeds [53]. In legumes, while free amino acids can increase markedly in concentration during germination, they are overall a small percentage of the total free and protein amino acids (Table 3). In *L. culinaris*, the total amount of free amino acids was 2.2 mg/g of dry weight, which increased to 48.6 mg/g on germination, and a substantial part of that increase is accounted for by asparagine alone (Table 3). Protein amino acids also increased during germination (with total amino acids increasing from 160 to 258 mg/g dry weight), with amino acids lysine and aspartate/asparagine increasing 4-fold (Table 3). Thus, the overall picture of change in free and total amino acid change is intriguing, with asparagine markedly increasing in both free and non-free amino acids during germination. To date, no clear hypothesis has been put forward to explain these observations. Further analysis is needed to distinguish whether the increase in non-free amino acids of asparagine is mainly seen in short peptides (such as dipeptides), which seems more likely, or in larger proteins.

Interestingly, while most amino acids increased as a proportion of dry weight during germination, the two sulfur-containing amino acids, cysteine and methionine, which are already extremely low in chickpea, appeared to decline further in concentration during germination (Table 3). This raises the possibility that the depletion of these amino acids, which may be intended to disrupt nutrition of potential herbivores, may also remain depleted in the seedling, where the possibility of fungal or insect predation is likely to be high.

While pea shows germination-related changes in free amino acids that are broadly similar to the pattern seen in lentil (Table 4), other species are less similar. The proportion of essential amino acids is relatively stable through chickpea germination (Figure 2). In the common bean, the increase in free asparagine is much more modest (Table 4), while free arginine and free glutamic acid are high pre-germination, which then drops after germination. These decreases mainly account for the overall decline in free amino acid concentration after germination in the common bean (48 mg/g to 33, Table 4). Thus, there are clearly marked differences among legume species with regards to changes in free amino acid content during germination.

## 3. Changes in Protein Digestibility during Legume Germination

A review by Sa et al. summarizes most of the in vitro and in vivo methods used to calculate protein digestibility [62]. In vivo methods include true digestibility, protein efficiency ratio (PER), protein digestibility-corrected amino acid score (PDCAAS) and digestible indispensable amino acid score (DIAAS). PER is calculated by feeding a test protein diet and casein to rats and then calculating the ratio of weight gain and the amount of protein consumed. PDCAAS is calculated as mg (limiting amino acid in 1 g of test protein)/mg (same amino acid in 1 g of reference protein), multiplied by the fecal true digestibility percentage [63]. The limiting amino acid is the essential amino acid present in the lowest proportion as compared with the reference to a food protein such as egg white. Fecal true digestibility is calculated as the difference in percentage of ingested and excreted amount of nitrogen. In vitro protein digestibility (IVPD) mimics the digestive process occurring in the gastrointestinal tract. It calculates the percentage of protein hydrolyzed in the presence of digestive enzymes [64].

For all species shown in Table 5, except for lentil, there was a moderate increase in in vitro protein digestibility after germination. For some species, this corresponded to almost an additional one-fifth of protein being digestible. The increase was seen gradually over chickpea germination, with IVPD of 68% in the seed; 70% after soaking; and 72%, 76% and 79% after 3-, 4- and 5-day germination, respectively [11].

A more advanced approach to assessing the bioavailability of protein breakdown products during intestinal digestion includes not only treatment of foods with the luminal stomach and pancreatic enzymes, but also over 20 brush border exopeptidase and endopeptidase enzymes, which play a key role in releasing amino acids, dipeptides, and tripeptides for transfer across the gut barrier [65]. A recent study contrasted raw and sprouted chickpeas after their passage through oral, stomach, duodenal and brush border digestion phases, in order to identify peptides that were resistant to all phases of digestion [66]. They noted a doubling of free alpha-amino nitrogen after sprouting, reflecting an increase in amino acids and oligopeptides. A drop in the number of larger peptides detectable by MS/MS after sprouting indicated that germination decreased the number of peptides that were resistant to digestion by all four digestion phases [66].

## 4. Changes in Proteases during Legume Germination

Proteases which break down seed proteins should ideally have reasonable efficiency at digesting their major substrates, and should be either localized in or delivered to the protein bodies where storage proteins are located [75]. Proteases are either endopeptidases or exopeptidases. Exopeptidases may be either aminopeptidases or carboxypeptidases.

The major classes of proteases are serine proteases, cysteine proteases, aspartic proteases, and metalloproteases, named according to their key enzymatic active site residues [76,77]. Serine proteases are broadly classified as trypsin-like and subtilisin-like based on their structures. Cysteine proteases are most effective at pH 4–6.5, such as papain. Metalloproteases require divalent metal cations such as Zn^2+^, Mg^2+^ or Ca^2+^.

In mung bean, chickpea, cowpea, and lentils with up to 3 days of germination [77], endoprotease activity was measured with a casein substrate, and shown to generally increase with germination time, especially in chickpea. Mung bean endopeptidase and exopeptidase activities increased on germination: endopeptidase activity started increasing after the third day, with a 10-to-15-fold increase by day 6; while the carboxypeptidase activity increased by 50% over the 6 days [78].

It remains to be elucidated to what extent the increased enzymatic activities during germination can be accounted for by reductions in protease inhibitors (both proteinaceous and others), by alterations in physiological conditions favoring more efficient enzyme function, by cleavage activation of pro-enzymes into active form, and by increased de novo synthesis of proteases.

## 5. Changes in Protease Inhibitors during Legume Germination

Protease inhibitor proteins constitute about 10% of the total protein content and are present in a high percentage of legume seed proteins [79]. The two main groups of legume protease inhibitor groups are Bowman–Birk inhibitors (BBI) and Kunitz-type inhibitors, the concentrations of which vary with species [80,81]. BBIs are 8–10 kDa double-headed serine protease inhibitors of ~71 amino acids in length that contain seven disulphide bonds (Figure 3). They have two active sites and inhibit both trypsin and chymotrypsin proteases. Kunitz-type inhibitors are between 8–22 kDa and have two disulphide bonds and one active site. They reversibly bind to serine, cysteine, and aspartic acid proteases, forming stable complexes that can inhibit competitively or non-competitively [82]. Protease inhibitors have inhibitory actions against plant pathogen by antinutritional interactions [83]. They can also cause hyperproduction of digestive enzymes, which results in loss of sulfur-containing amino acids, weakening the insects and finally causing their death [84]. Different classes of pests utilize different digestive enzymes, with some using cysteine proteases while other use serine proteases [83]. The mechanism by which the plant protease inhibitors bind to the insect proteases is similar for all the four classes of inhibitors: aspartic acid protease inhibitors (pepstatins), serine protease inhibitors (serpins), cysteine protease inhibitors (cystatins), and metallo carboxy protease inhibitors [83]. The specificity of the inhibitor–protease interaction depends upon the specificity of proteolytic activity of the proteases [84]. Many naturally occurring protease inhibitors from plants, including legumes, can have also effects on humans [80]. Protease inhibitors severely impact the proteolytic activity in gastrointestinal tract, thereby limiting the nutrients absorption and digestibility [85]. Chickpea, lentil and pea Bowman–Birk inhibitors inhibit in vitro cancer cell growth [84,86,87].

Trypsin inhibitory activity in various legumes typically reduces, but is by no means eliminated, during germination (Table 6). Germination for 48 h lowered trypsin inhibitory activity by 64% in fava bean [88]. In lentils, trypsin inhibitory activity does not change much in the first 3 days of germination but decreases by up to 18% after 6 days of germination and by up to 45% after 10 days, which may help in supply of new amino acids for growing seedling [89]. In kidney beans, decreases in inhibitor content were observed only after 10 days of germination [90,91]. In cowpea, there was a 19% reduction of trypsin inhibitors after 8 days’ germination [69].

In addition to the protein-based protease inhibitors, non-protein components also interfere with protein digestion. Firstly, the physical barriers of cell wall and seed coat structures can hinder the proteolysis process [92], and these will undergo changes during germination. Secondly, the glycoproteins contained within the cell wall are typically unlikely to be easily hydrolyzed [93]. Thirdly, non-protein anti-nutritional factors (ANFs) include polyphenols, tannins, phytates, lectins and non-starch polysaccharides [62]. While they are likely present in order to deter herbivores (insects or other species), they are likely also to impact on the proteases of the germinating seeds themselves. Non-starch polysaccharides adsorb amino acids and peptides released during protein hydrolysis [62]. Phytates chelate various minerals, such as calcium and zinc, which are essential cofactors for many digestive enzymes. Phytates thus reduce protein digestibility, but their amounts can be reduced slightly during legume germination by increased expression of phytases [94,95]. A feasible way of increasing phytase activity in legume food preparations is to provide it from cereal grains such as wheat [95], where phytase is much more abundant. Increasing phytase activity is considered in animal feeds, primarily to aid bioavailability of minerals and trace elements [95], rather than reflecting concerns about the effects of phytates on protein digestion. Tannins also serve to reduce protein digestibility in legumes, as well as exerting other effects; however, their concentration can be reduced [96] by simple techniques, such as soaking in water and discarding the soaking broth. Other antinutrients that inhibit digestibility include lectins, whose effects can also be reduced by germination [96].

## 6. Changes in Protein Allergens during Germination

Various nutritionally important foods cause allergic reactions when consumed [101]. Among legumes, peanut is the source of the highest number of allergic proteins, followed by soybean, lentil, chickpea, pea, mung bean, pigeon pea, and lupin in decreasing order of allergenicity [102,103]. The common characteristics of legume allergic proteins are that they are often heat stable, stable to gastrointestinal fluids, water soluble, and glycosylated. They range in molecular weight between 10 to 400 kDa, and are classified into six families, namely cupins, prolamins, pathogenesis related (PR) proteins, profilins, vicilins and glycins. Of these, cupins and prolamins are seed storage proteins. The properties of various allergic protein families are summarized in Table 7 [101,102,103,104].

Germination results in decreased allergenicity of legumes. Studies by Troszyńska et al. (2007) have shown that germination decreases the immunoreactivity of peas by 40% and that of soybeans by 70%. Germination in darkness reduced the immunoreactivity of soybeans by 78% [105]. The electrophoresis bands of pea proteins showed the disappearance of molecular weight of 40 kDa fractions and fractions with higher molecular weight in germinated peas. However, vicilin retains its stable structure even after germination and is not hydrolyzed by proteases and thus contributes to the remaining allergenicity even after pea germination. Germination decreased immunoreactivity more significantly for soybeans than for pea [106], possibly because one of the major soy allergens, glycinin, has been shown to be hydrolyzed after 3 days of germination [106].

## 7. Impact of Food Processing on Protein Digestibility

Other than germination, various methods, such as cooking, extrusion, autoclaving, irradiation, roasting and fermentation, have helped in increasing the digestibility of plant proteins, presumably by deactivating the anti-nutritional factors such as protease inhibitors. They can also improve food flavor and texture. Food processing methods, such as autoclaving, cooking, and fermentation, moderately increase the in vitro protein digestibility (IVPD) of legumes [62,107]. Aviles-Gaxiola et al. classified the various methods to overcome trypsin-inhibitor activity into physical processes (thermal, extrusion, radiation, ultrasound), chemical processes (reducing agents, acids and bases, functionalized polymers), and biological processes (germination, fermentation) [108]. They compared various methods and found that, in soybean, thermal treatment along with reducing agent metabisulfite was best for reducing trypsin inhibitors, whereas, in chickpea, reducing agent L-cysteine was most effective. Trypsin inhibitors can be reduced by heating at a high temperature and prolonged boiling. Lectins can also be reduced by heat treatments. Phytates can be reduced by soaking in water or fermentation.

While crude and soluble protein content in soymeal increases with fungal or bacterial fermentation [109,110], there is an increase in in vitro protein digestibility from 60.5% to 67% with fungal fermentation and to 76% by bacterial fermentation, associated with increased essential amino acids and decreased allergenicity [111,112]. Fermentation of soybean with *Bacillus subtilis* increased the in vitro gastrointestinal digestibility by 1.52-fold, increased the amount of soluble peptides, and increased essential amino acids by ∼4%, especially arginine, tyrosine, histidine and phenylalanine [113]. Trypsin inhibitor activity decreased by more than 10-fold from 27.33 TIU/g in the unfermented soya meal to 2.14 TIU/g in *Bacillus natto-*fermented soymeal [114].

To date, there is little information regarding the effects of combining germination with other food processing techniques that may improve the digestibility of protein. Germination of the legume *Vigna unguiculata* L. for 96 h followed by autoclaving improves protein digestibility, in addition to completely destroying some antinutritional factors [69]. It will be of great interest to explore combinations of the above treatments with germination, to see how these combinations may significantly impact on protein digestibility.

## 8. Conclusions

Legumes are good sources of protein and represent valuable potential substitutes for animal protein production. This increases the impetus to better understand, modify and improve the digestibility and amino acid content of plant protein sources, and their changes during germination. The germination proteolysis program in legumes represents the culmination of selective forces over millions of years, where seeds are highly adapted to mechanisms of seed dispersion, seed dormancy, predator and pathogen repulsion, and also adapted for the efficient delivery of protein building blocks to the growing seedling. Layered on top of this are the effects of human selection for desirable traits in legume seeds in terms of yield, dietary quality, and storage properties. A number of overall features emerge from the literature reviewed here. Legume germination increases the amount of protein, increases the amount of soluble proteins, breaks down high molecular weight polypeptides, increases proteolytic activity, reduces protease inhibitors, increases mammalian digestibility, and reduces the amount of allergenic proteins (Figure 4). In some legumes, there is a marked increase of certain free amino acids during germination, particularly asparagine, although the great majority of amino acids in germinated legumes remain bound up in proteins.

Improved knowledge of changes in the legume seed proteome and its changes during germination and other conditions may help guide improvement of their nutritional and other properties [115,116,117]. Mass spectrometry-based proteomics and peptidomics of seeds are versatile tools which can track changes in protein and peptide distributions in seeds during germination [118]. Shotgun proteomics can profile germination-related changes in storage proteins, proteases, and protease inhibitors. Peptidomics can identify changes in small peptides and their termini. This has the potential to identify changes in allergenic peptides and in potentially bioactive peptides [119], and to characterize the proteolytic activities that dominate during germination. Application of these emerging technologies, alongside established food chemistry approaches, has the potential to rapidly uncover and quantify many of the unknown changes that occur during the program of protein breakdown during germination.

Given the effects of different food processing techniques on protein digestibility, it is of great interest to explore the combined effects of germination with heating, fermentation, and other approaches, to identify whether further improvements in digestibility may result. Natural food processing techniques, such as heating, germinating, and fermenting, of legumes [120] or other plant protein sources may be combined in different ways to determine which combinations provide the most effective increase in digestibility. It will be of great interest to determine which traditional existing food preparation techniques [121] may have evolved to have such beneficial effects, and to what extent a more scientifically-driven assessment of digestibility of different novel food preparation approaches can unlock additional digestibility.

## Figures and Tables

**Figure 1 molecules-28-03204-f001:**
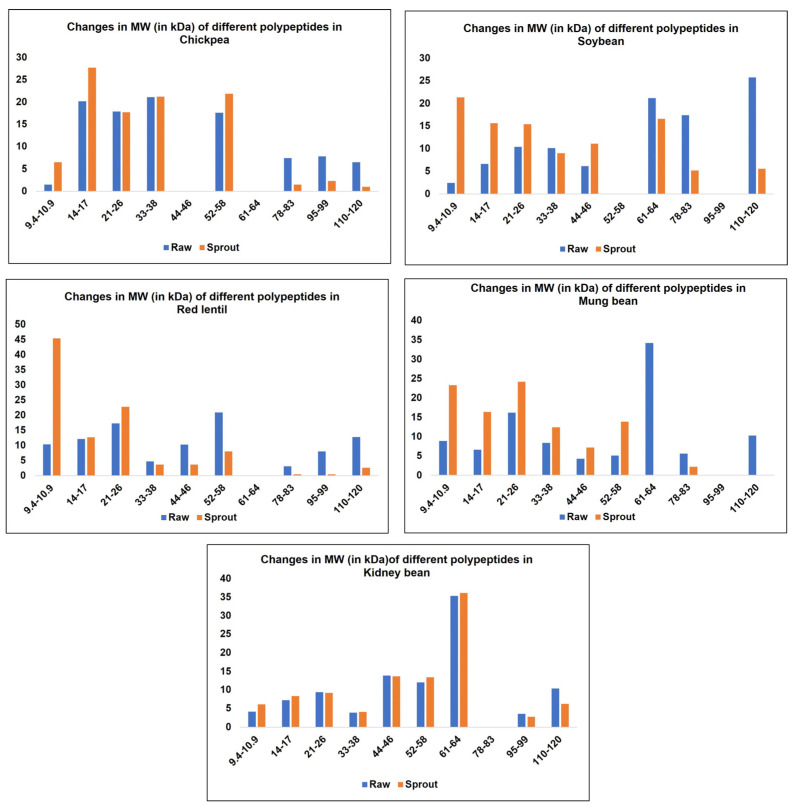
Changes in molecular weight of different polypeptides after germination at 40 °C in various legumes (% of total protein) (adapted with permission from Mammila et al, Effect of germination on antioxidant and ACE inhibitory activities of legumes; published by Elsevier, 2017) [49].

**Figure 2 molecules-28-03204-f002:**
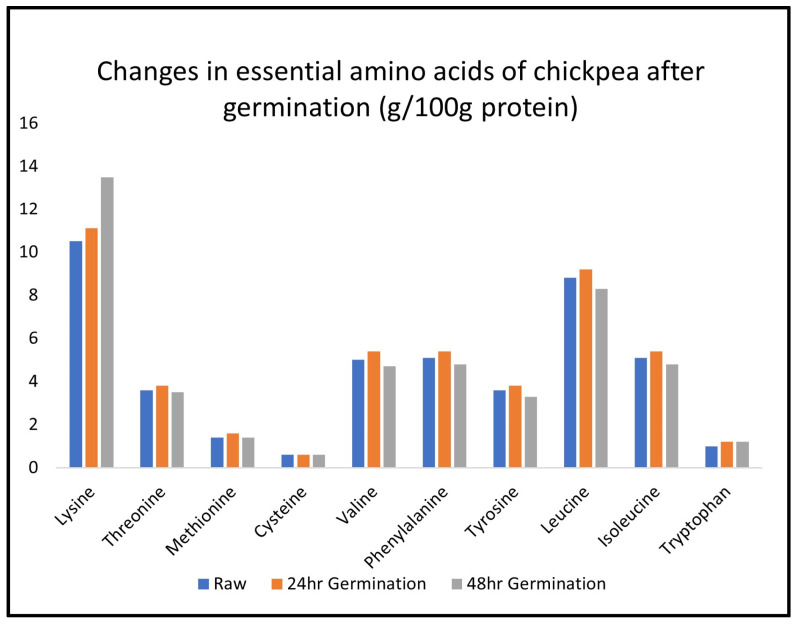
Changes in essential amino acid content in chickpea after germination [61].

**Figure 3 molecules-28-03204-f003:**
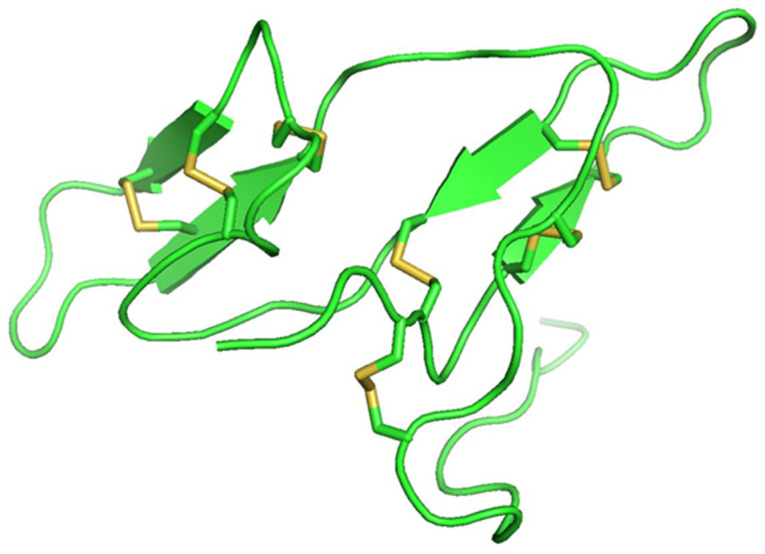
Bowman–Birk inhibitor from pea (PDB ID: 1pbi) showing seven disulfide bonds (yellow).

**Figure 4 molecules-28-03204-f004:**
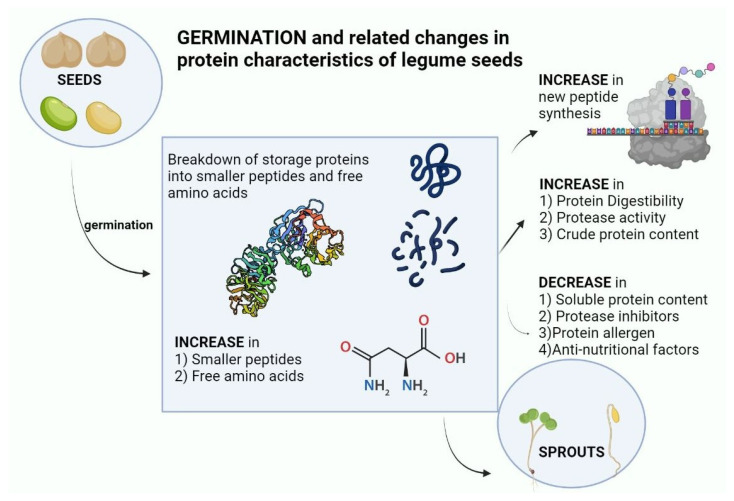
Changes in protein characteristics of legume seeds on germination (created with BioRender.com URL (accessed on 18 January 2023).

**Table 1 molecules-28-03204-t001:** Changes in crude protein content after germination (calculated using Kjeldahl method and expressed as a percentage of dry weight, unless stated). Non-legume comparison species are shown in brackets.

Sprout Species	Pre-Germination	Post-Germination	References
Chickpea	24.4%	27.7%	Xu et al. (2019) [33]
	18.4%	24.6%	Ferreira et al. (2019) [34]
	32 ± 1.8% *	48 ± 0.5% *	Dipnaik and Bathere (2017) [35]
	22.3%	24.1%	Mansour (1987) [36]
	~20%	23.9%	Khalil et al. (2007) [37]
	20.3%	23.6%	Uppal et al. (2012) [11]
Chickpea desi	14.8 ± 0.6%	15.9 ± 0.4%	Kumar et al. (2019) [38]
Chickpea desi	~21%	24.1%	Khalil et al. (2007) [37]
Mungbean	22.5 ± 0.9%	36 ± 0.5%	Dipnaik and Bathere (2017) [35]
	22.3%	24.9%	Uppal et al. (2012) [11]
Cowpea	30 ± 1.07%	40 ± 0.5%	Dipnaik and Bathere (2017) [35]
	22.5%	24.9%	Uppal et al. (2012) [11]
Moth bean	30 ± 1.0%	40 ± 12.3%	Dipnaik and Bathere (2017) [35]
Soybean	40.2 ± 0.3%	46.3 ± 0.4%	Joshi and Varma (2016) [39]
	39.1%	45.1%	Kayembe et al. (2013) [40]
Faba bean	26.4%	30.6%	Kassegn et al. (2018) [41]
Pea			
var *ucero*	25.4 ± 0.1%	27.0 ± 0.1%	Martinez-Villaluenga et al. (2008) [42]
var *ramrod*	21.1 ± 0.0%	22.7 ± 0.1%	Martınez-Villaluenga et al. (2008) [42]
var *agra*	22.9 ± 0.1%	22.7 ± 0.1%	Martınez-Villaluenga et al. (2008) [42]
Black gram	20 ± 1.5%	36 ± 1.54%	Dipnaik and Bathere (2017) [35]
Pigeon Pea	19.53 ± 0.02%	22.54 ± 0.02%	Rizvi et al. (2022) [43]
Black soybean flour	39.46 ± 0.08%	43.30 ± 0.05%	Mitharwal and Chauhan (2022) [44]
Chickpea flour	21.9 ± 0.2%	24.0 ± 0.2%	Sofi et al. (2023) [45]
Pigeon pea flour	22.71 ± 0.15%	26.72 ± 0.11%	Chinma et al. (2022) [46]
(Broccoli)	26.1%	29.8%	Taraseviciene et al. (2009) [47]
(Brown Rice)	6.9 ± 0.0%	8.9 ± 0.2%	Moongngarm et al. (2010) [48]

* Calculated using Biuret method.

**Table 2 molecules-28-03204-t002:** Changes in soluble protein fractions (percentage of total protein) after germination.

Sprout Species	Pre- Germination	Post- Germination	References
Chickpea (major globulin)	45.85%	37.08%	Portari et al. (2005) [56]
Pea (albumin and globulin)			Martinez-Villaluenga et al. (2008) [42]
var *ucero*	28.95%	24.9%	
var *ramrod*	26.71%	22.69%	
var *agra*	26.67%	25.03%	
Lupin	12.81%	15.7%	Villacrés et al. (2015) [57]
Sweet lupin (albumin and globulin)			Gulewicz et al. (2008) [42]
*Lupinus luteus* cv. 4486	36.89%	39.45%	
*Lupinus luteus* cv. 4492	39.63%	35.91%	
*Lupinus angustifolius* cv.troll	34.9%	35.2%	
*Lupinus angustifolius* cv.zapato	35.4%	29.48%	
(Sorghum)	25%	28%	Afify et al. (2012) [58]

**Table 3 molecules-28-03204-t003:** Changes in free amino acids and protein amino acids (mg/g dry weight) in lentil, *Lens culinaris*, before and after germination [59].

Free Amino Acids	Pre- Germination	Post- Germination	All Amino Acids	Pre- Germination	Post- Germination
**Arg**	0.1	0.94	Arg	10.61	12.11
**His**	0.22	0.68	His	8.74	10.79
**Ile**	0	2.06	Ile	6.26	11.44
**Leu**	0	2.05	Leu	10.64	17.1
**Lys**	0	0.93	Lys	4.54	16.99
**Met**	0	0.26	Met	1.49	1.02
**Cys**	0	0	Cys	0.4	0
**Phe**	0	2.32	Phe	6.7	11.56
**Tyr**	0	1.1	Tyr	6.34	7.7
**Pro**	0.17	3.24	Pro	11.11	10.84
**Ser**	0	2.64	Ser	11.38	15.54
**Thr**	0.03	1.18	Thr	5.57	7.14
**Val**	0	2.83	Val	8.54	13.23
**Trp**	0	0.52	Trp	0	0
**Ala**	0.45	3.21	Ala	20.42	36.76
**Asp**	0.18	0.32	ASX *	10.96	41.39
**Asn**	0.51	18.96			
**Glu**	0.48	3.15	GLX *	26.55	34.12
**Gln**	0	1			
**Gly**	0.05	1.23	Gly	9.77	10.77
**Total**	2.19	48.62		160.02	258.50

**Table 4 molecules-28-03204-t004:** Germination-related changes in legume free amino acids (in mg/g dry weight) [60].

Amino Acids	Beans (*Phaseolus Vulgaris*)		Lentils (*Lens Culinaris*)		Pea (*Pisum Sativum*)	
Germination	Pre-	Post-	Pre-	Post-	Pre-	Post-
Alanine	2.93	4.4	0.5	0.8	0.25	2.58
Arginine	13.2	2.95	0.6	1.2	3.4	3.8
Asparagine	5.9	8.0	0.88	28.7	0.8	23.0
Aspartic acid	4.0	2.5	0.70	1.27	1.6	5.4
Glutamic acid	11.2	4.09	1.34	3.93	2.0	3.5
Glutamine	0.0	1.27	0.0	1.06	0.0	2.38
Glycine	0.50	0.09	0.08	0.35	0.07	0.30
Histidine	0.45	0	0.04	0.80	0.14	0.0
Isoleucine	0.8	0.8	0.0	0.99	0.0	0.58
Leucine	0.6	1.2	0.0	0.61	0.01	0.52
Lysine	0.11	0.53	0.0	1.05	0.08	0.75
Methionine	0.0	0.382	0.0	0.0	0.0	0.0
Phenylalanine	1.0	0.8	0.0	1.04	0.15	1.16
Proline	0.8	0.8	0.23	2.91	0.53	2.23
Serine	0.1	2.0	0.0	2.43	0.02	1.51
Threonine	0.2	1.0	0.042	2.12	0.0	0.36
Tryptophan	0.68	0.33	0.0	0.27	0.10	0.50
Tyrosine	4.0	0.33	0	0.64	0.06	0.52
Valine	2.0	1.8	0.11	2.42	0.0	1.78
Total	48.47	33.27	4.52	52.59	9.2	50.86

**Table 5 molecules-28-03204-t005:** Changes in vitro protein digestibility (IVPD) after germination.

Sprout Species	Pre-Germination	Post-Germination	References
Chickpea	67.7% 75.4% 64.2 ± 1.8%	79.0% 86.5% 73.4 ± 0.7%	Uppal et al. (2012) [11] Chitra et al. (1996) [67] Ghavidel et al. (2007) [68]
Chickpea flour Mungbean	83.8 ± 2.8% 66.4% 70.9%	88.5 ± 3.2% 83.0% 82.7%	Sofi et al. (2023) [45] Uppal et al. (2012) [11] Chitra et al. (1996) [67]
Cowpea	73.3% 71.2 ± 0.1%	85.7% 73.5 ± 0.4%	Uppal et al. (2012) [11] Kalpanadevi et al. (2013) [69]
Soybean	63.3%	73.6%	Chitra et al. (1996) [67]
Pigeon pea	69.1%	85.1%	Chitra et al. (1996) [67]
Pigeon pea flour Kidney bean	72.30 ± 0.24% 80.6 ± 0.02%	82.66 ± 0.17% 87.1 ± 0.03%	Chinma et al. (2022) [46] Shimelis et al. (2006) [70]
Yellow pea	78.6 ± 0.1%	79.9 ± 0.1%	Setia et al. (2019) [71]
Fava bean	78.0 ± 0.2%	80.4 ± 0.1%	Setia et al. (2019) [71]
Lentil	65.6 ± 1.1%	64.2 ± 1.8%	Ghavidel et al. (2007) [68]
Pigeon Pea Lupin Green gram	68% 73.0 ± 4.87% 61.0 ± 1.0%	88% 74.3 ± 1.89% 72.7 ± 0.8%	Rizvi et (2022) [43] Munoz-Landes et al. (2022) [72] Ghavidel et al. (2007) [68]
Soy milk (Sorghum)	80% 51%	85% 65%	Hu et al. (2022) [73] Afify et al. (2012) [58]
(Red sorghum)	48%	68.1%	Onyango et al. (2013) [74]
(Pearl millet)	21.5%	34.5%	Onyango et al. (2013) [74]

**Table 6 molecules-28-03204-t006:** Changes in trypsin inhibitors (in trypsin-inhibitory units/mg of protein, unless mentioned) after germination for 72 h.

Sprout Species	Pre-Germination	Post-Germination	References
Chickpea	11.9	7.86	El-Adawy (2002) [97]
Mungbean	16.5	12.8	El-Adawy et al. (2003) [97]
Pea	10.8	8.6	El-Adawy et al. (2003) [97]
Lentil	33.3	27.3	El-Adawy et al. (2003) [97]
Horsegram	11.5	8.4	Pal et al. (2013) [98]
Kidney bean			Shimelis et al. [74]
*Roba variety*	4.5	3.8	
*Awash variety*	20.8	17.3	
*Beshbesh variety*	29.2	24.5	
French bean	3.1	2.2	Alonso et al. (1999) [99]
Fava bean	4.4	3.3	Alonso et al. (1999) [99]
Soybean	275 mg/g	225 mg/g	Wu et al. (2023) [100]
(Sorghum)			
*Hamra* variety	31.6	19.9	Osman et al. (2013) [92]

**Table 7 molecules-28-03204-t007:** Major allergic protein families of legumes and their significant characteristics.

Allergic Protein Family	Characteristics
**Prolamin superfamily**	Largest family of plant food allergens, low molecular weight, sulfur-rich, glycosylated, includes 2S storage proteins from legumes, non-specific lipid transfer proteins, protease inhibitors
**Cupin superfamily**	Consists of two conserved consensus sequence motifs, β barrel structural domain, seed storage proteins of soybeans and peanuts
**Pathogenesis-related proteins**	Comprised of 14 different unrelated protein families, small size, stable in acidic conditions, Increased synthesis during environmental and pathogen stresses
**Profilins**	Small 12–15 kDa MW, highly conserved sequences, cytoplasmic immunological cross-reactivity with pollens
**Vicilins**	Part of the globulin family, anti-fungal, anti-microbial activity
**Glycilins**	Hexamer, 300–400 kDa

## Data Availability

Not applicable.

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
