# Peer review of "Relationship between Protein Digestibility and the Proteolysis of Legume Proteins during Seed Germination"

_molecules, 2023, doi:10.3390/molecules28073204_

Round 1

Reviewer 1 Report

I have gone through the manuscript entitled "Relationship between protein digestibility and the proteolysis of legume proteins during seed germination". In this review the authors have discussed the different kinds of proteins and amino acids  involved during seed germination. Species specific proteins and amino acids their significant digestibility of sprouts of various legumes. The manuscript is important researchers especially who are involved in legume proteins and their role in germination processes. However I suggest the authors to write the importance of legumes in introduction section highlighting their food value. I also suggest to update the reference including the latest reference viz.,8. Jha, U.C.; Nayyar, H.; Parida, S.K.; Deshmukh, R.; von Wettberg, E.J.B.; Siddique, K.H.M. Ensuring Global Food Security by Improving Protein Content in Major Grain Legumes Using Breeding and ‘Omics’ Tools. Int. J. Mol. Sci. 2022, 23, 7710.

There are some minor correction required.

(Phaseolus vulgaris, Vigna unguiculata, Phaseolus lunatus, Vicia faba), chickpea (Cicer  arietinum), lentil (Lens culinaris) and lupin (Lupinus albus) .The name of these legumes will be in italics.

Line no. 141 In L culinaris, the it will be In L. culinaris, 

 Line no. 183 changes in protein digestibility during legume germination 

Changes will be in capital letter

Author Response

However I suggest the authors to write the importance of legumes in introduction section highlighting their food value. I also suggest to update the reference including the latest reference viz.,8. Jha, U.C.; Nayyar, H.; Parida, S.K.; Deshmukh, R.; von Wettberg, E.J.B.; Siddique, K.H.M. Ensuring Global Food Security by Improving Protein Content in Major Grain Legumes Using Breeding and ‘Omics’ Tools. Int. J. Mol. Sci. 2022, 23, 7710.
We have added the importance of legumes in the introduction section including this reference (see lines 47-54).

2. (Phaseolus vulgaris, Vigna unguiculata, Phaseolus lunatus, Vicia faba), chickpea (Cicer arietinum), lentil (Lens culinaris) and lupin (Lupinus albus) .The name of these legumes will be in italics.
Done (This change has been done in lines 44-46 ).

3. Line no. 141 In L culinaris, the it will be In L. culinaris,
Done

4. Line no. 183 (187) changes in protein digestibility during legume germination,
change in capital
Done

Reviewer 2 Report

Dear authors,

Please consider these suggestions and corrections, before publishing the article. Kind regards.

Review:

-          Line 12-13 Legume seed protein is an important source of nutrition, but it is less digestible than ani- 12 mal protein. Add generally to it, because it is a broad statement.

-          Scientific names at their first appearance in the text must be followed by the name of the person who identified them.  (e.g., Pisum sativum L.);

-          Line 56 – Maybe use another term for “release the protein”.

-          Line 65 – Cite which conditions are “proper physiological conditions”.

-          Line 96 – Add something about the change in N content during the germination, as N is very associated with protein. Does this increase in protein occur due to N uptake or due to modifications in the N use of the seeds? Or a combination.

-          Table 1. These results based on the Kjeldahl method, may not express the true protein. Many compounds have N but are not true proteins (NH3, Amino acids, nucleic acid). This must be stated in the review, see the previous comment about N dynamics.

-          Tables and figures can be improved in terms of design: remove figure contours, and lines from the middle of the table.  

-          In the paragraph between lines 265-283, please add some notes about proteins bound to the cell walls (indigestible protein).

Author Response

1. Line 12-13 Legume seed protein is an important source of nutrition, but it is less digestible than ani- 12 mal protein. Add generally to it, because it is a broad statement.
We have added “generally” to the abstract first line. ( line number 14)

2. Scientific names at their first appearance in the text must be followed by the name of the person who identified them. (e.g., Pisum sativum L.);
We have added the names of scientists who have identified the particular species (lines 44-46).

3.Line 56 – Maybe use another term for “release the protein”.
We have changed the term and rephrased the sentence (lines 60-61)

4.Line 65 – Cite which conditions are “proper physiological conditions”.
We have added the conditions required for germination in lines 75-76.

5. Line 96 – Add something about the change in N content during the germination, as N is very associated with protein. Does this increase in protein occur due to N uptake or due to modifications in the N use of the seeds? Or a combination.

6.Table 1. These results based on the Kjeldahl method, may not express the true protein. Many compounds have N but are not true proteins (NH3, Amino acids, nucleic acid). This must be stated in the review, see the previous comment about N dynamics.
We thank the reviewer for highlighting this important point. We have addressed these two related queries in line 104 to 111.

7. Tables and figures can be improved in terms of design: remove figure contours, and lines from the middle of the table.
We have made changes to both figures and tables, in order to improve the layout and presentation.

8. In the paragraph between lines 265-283, please add some notes about proteins bound to the cell walls (indigestible protein).
We have addressed this issue in lines 300-301.

Reviewer 3 Report

1.     The Latin names of legumes should be written in italics.

2.     Broccoli and brown rice in Table 1, sorghum in Table 2 and Table 7, red sorghum and pearl millet in Table 5 are not legumes.

3.     The section of ‘Impact of food processing on protein digestibility’ does not seem to be directly related to the topic of this article which is germination, or this section does not further explore the combined effects of other food processing methods and germination.

4.     It is suggested to discuss whether the time and other conditions have significant effects on the seed germination.

5.     There are few references published in recent years.

6.     The graphs in this manuscript are not well drawn, they need to be improved.

7.     There are some spelling mistakes, such as extra spaces, etc.

Author Response

Comments and Suggestions for Authors

1. The Latin names of legumes should be written in italics.

Done

2. Broccoli and brown rice in Table 1, sorghum in Table 2 and Table 7, red sorghum and pearl millet in Table 5 are not legumes.

We agree that this was confusing. We have indicated in Table 1 that these values are presented for comparison only.

3.The section of ‘Impact of food processing on protein digestibility’ does not seem to be directly related to the topic of this article which is germination, or this section does not further explore the combined effects of other food processing methods and germination.

We have improved this section by including reference to a study where germination was combined with autoclaving, resulting in increased protein digestibility. Lines 372-379

4. It is suggested to discuss whether the time and other conditions have significant effects on the seed germination.

We have added the important physiological conditions for germination in the lines 75-76

5. There are few references published in recent years.

The tables have been updated with more recent references (highlighted in yellow). Additional recent references have been added in lines 224 onwards, and lines 224-234 and 401-402.

6. The graphs in this manuscript are not well drawn, they need to be improved.
We have corrected the tables and the figures.

7. There are some spelling mistakes, such as extra spaces, etc.
These have been corrected.

Round 2

Reviewer 1 Report

Authors have revised the manuscript and it can be accepted.

Reviewer 3 Report

The revision can be accepted.